# The Pigment World: Life’s Origins as Photon-Dissipating Pigments

**DOI:** 10.3390/life14070912

**Published:** 2024-07-22

**Authors:** Karo Michaelian

**Affiliations:** Department of Nuclear Physics and Application of Radiation, Instituto de Física, Universidad Nacional Autónoma de México, Circuito Interior de la Investigación Científica, Cuidad Universitaria, Cuidad de México CP 04510, Mexico; karo@fisica.unam.mx

**Keywords:** pigment world, fundamental molecules, origin of life, evolution, dissipative structuring, entropy production, prebiotic chemistry, abiogenesis, adenine, biosphere, natural selection, 92-10, 92C05, 92C15, 92C40, 92C45, 80Axx, 82Cxx

## Abstract

Many of the fundamental molecules of life share extraordinary pigment-like optical properties in the long-wavelength UV-C spectral region. These include strong photon absorption and rapid (sub-pico-second) dissipation of the induced electronic excitation energy into heat through peaked conical intersections. These properties have been attributed to a “natural selection” of molecules resistant to the dangerous UV-C light incident on Earth’s surface during the Archean. In contrast, the “thermodynamic dissipation theory for the origin of life” argues that, far from being detrimental, UV-C light was, in fact, the thermodynamic potential driving the dissipative structuring of life at its origin. The optical properties were thus the thermodynamic “design goals” of microscopic dissipative structuring of organic UV-C pigments, today known as the “fundamental molecules of life”, from common precursors under this light. This “UV-C Pigment World” evolved towards greater solar photon dissipation through more complex dissipative structuring pathways, eventually producing visible pigments to dissipate less energetic, but higher intensity, visible photons up to wavelengths of the “red edge”. The propagation and dispersal of organic pigments, catalyzed by animals, and their coupling with abiotic dissipative processes, such as the water cycle, culminated in the apex photon dissipative structure, today’s biosphere.

## 1. Introduction

Despite their seemingly spontaneous emergence, irreversible processes are contingent on the dissipation of a conserved quantity (e.g., energy, momentum, angular momentum, charge) from their environment. “Dissipation” in this thermodynamic context refers to the distribution of the conserved quantity over a greater number of microscopic degrees of freedom, known in thermodynamic terms as “entropy production”. Dissipation is irreversible, i.e., increasing only in the forward direction in time, thereby establishing the second law of thermodynamics. In the process, macroscopic structures can be formed, which promote the dissipation of the conserved quantity. The formalism to treat such “dissipative structuring” was developed by Prigogine and coworkers [1,2] and explains the seemingly spontaneous emergence of structures such as energy, material, or charge flows, which dissipate a temperature, concentration, or electrical potential, respectively; or more complex space and time symmetry breaking dissipative structures in non-linear systems such as convection cells, which dissipate energy over a temperature gradient, or the Belusov–Zhabotinsky reactor system density patterns, which dissipate an imposed chemical potential.

Irreversible thermodynamic theory, involving forces, flows, dissipative structures, and entropy production, provides an understanding of many “vital” characteristics seen in biology and its interaction with its environment [1,3,4,5,6,7,8,9,10,11,12,13]. Understanding the “spontaneous” abiogenesis of the irreversible process of life, therefore, requires identifying the conserved quantity (or quantities) dissipated and the corresponding primordial dissipative structures or processes formed to promote this dissipation.

The fact that the sun has been the most revered deity in past religions suggests that humans have always understood the singular importance of the sun to life. This historic intuition, taken together with a modern understanding of “dissipative structuring” [1,2] leads to a framework from within which the abiogenesis of life and its evolution can be understood. The “thermodynamic dissipation theory for the origin and evolution of life” [8,10,12,14] proposes that life is the irreversible process that arose in response to dissipating the incident UV-C solar light flux available at Earth’s surface throughout the Archean. This irreversible process consisted of the creation, through dissipative structuring, of organic chromophores (pigments) that could absorb these photons and dissipate the resulting electronic excitation energy into vibrational energy of the pigment and the surrounding water solvent molecules. This structuring of a “UV-C Pigment World” from simpler and common carbon-based precursor molecules occurred at the ocean surface under UV-C photons of between 205 to 320 nm wavelength (soft UV-C plus UV-B). These photons arrived at the Earth’s surface throughout the Archean with an important integrated intensity of ∼5 W m^−2^ [15,16], and each photon had sufficient energy to re-configure covalent bonds of carbon-based molecules, but not enough energy to excessively ionize and therefore dissociate the molecules.

The singular importance of pigments to the development of life is observable in the historical fossil record. Organic pigments are first seen to arise in the UV-C and then evolve gradually to cover the entire solar spectrum up to the red edge (∼700 nm) [17]. Contemporary biology, embracing Darwinian principles, recognizes pigments of phototrophic organisms as primary, or accessory, light collection molecules or as photo-protective agents of the photosynthetic system [18]. However, broadband absorption of photons over the entire spectrum (Figure 1) is not congruent with the tenants of Darwinian theory since broadband absorption and rapid dissipation of the electronic excitation energy into vibrational energy (heat) requires numerous complex pigments and their support structures. Transparency, or reflectivity, to the supposedly “offending” light, while optimizing absorption only at wavelengths directly involved in photosynthesis (∼680 nm), would seem to be a more efficient use of resources.

Although pigments screening certain UV wavelengths may indeed be protective in contemporary photosynthetic organisms incorporating weak Van der Waals bonding, screening the visible part of the spectrum (e.g., the secondary carotenoids) remains enigmatic [18]. Furthermore, even though isolated chlorophylls are susceptible to bleaching (the opening up of the tetrapyrrole ring) at UV-A wavelengths (350 nm) where there is still significant absorption, accessory pigments have rather little effect on chlorophyll bleaching, independent of the wavelength of the incident light [19]. Their assignation as protectors of the photosynthetic system is, therefore, probably misguided. Interestingly, high concentrations of chlorophylls form aggregates that are particularly rapid quenchers from the UV-C to the UV-A and are, therefore, efficient and stable dissipative structures [19].

**Figure 1 life-14-00912-f001:**
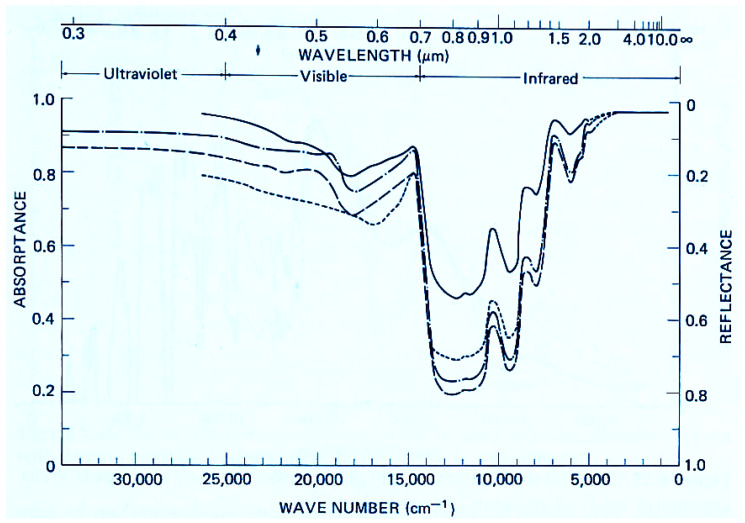
The wavelength dependence of photon absorption on the leafs of 4 contemporary plant species. Absorption is strong over the solar spectrum, from the UV-C at about 205 nm to the red edge at 700 nm (0.7 μm). Photosynthetic efficiency, however, is usually less than 1%, with useful photons being limited to the chlorophyll peak at around 680 nm. The small dip in the absorption spectrum at 550 nm gives plants their green color. Reprinted with permission from Gates [20].

Less than 1% of free energy in light absorbed by today’s photosynthetic organisms is converted into chemical potential energy through photosynthesis [21]. The rest is dissipated into heat. In only a few generations, humans have been able to double the photosynthetic efficiency of crops through artificial selection [21], suggesting that nature, with over 500 million years of natural selection operating on plants, has not been overly concerned with photosynthetic efficiency. The thermodynamic dissipation theory, instead, argues that natural thermodynamic selection on optimization of dissipation of the incident solar spectrum into heat, rather than photosynthetic efficiency (or reproductive success), is the true object of natural selection and leads to the evolution of organic pigments and biotic organisms and the biosphere [8,14,17]. Supporting this conjecture is the fact that organic material dissipates the solar spectrum at Earth’s surface significantly more efficiently than does inorganic material, particularly towards shorter wavelengths [22] where photon dissipation gives greater entropy production.

Subsequent evolution of dissipative structuring occurred towards the production of pigments dissipating the higher intensity, but lower energy, visible photons of wavelengths up to the red edge. Today, all pigments, including the original UV-C pigments, are dissipatively structured through complex biosynthetic pathways using visible light. Information for building these pathways became programmed into the DNA of organisms. Information stored in the collective gene pool of all organisms is relevant today for photon dissipation at the biosphere level, now dissipating the high intensity near UV and visible wavelengths, with dissipation of the lower intensity UV-C relegated to the life-derived chromophores oxygen and ozone in the stratosphere.

The aim of this paper is to provide empirical evidence, and propose physical-chemical mechanisms, for understanding the abiogenesis of life as a thermodynamically irreversible process arising through the dissipative structuring of an organic “Pigment World” to dissipate first the UV-C surface light of the Archean, and, later, visible light. The first pigments in the UV-C are known today as the “fundamental molecules of life”, i.e., those molecules at the foundations of life and common to all three domains: archaea, bacteria, and eukaryotes.

## 2. The Fundamental Molecules of Life Are Dissipatively Structured UV-C Pigments

### 2.1. Precursors of the Fundamental Pigments of Life

The primordial atmosphere of Earth was most likely removed by the “late heavy bombardment” occurring around 3.9 Ga [23]. Earth’s secondary atmosphere, which provided precursors for the fundamental molecules of life, was made up of the common volcanic gases H_2_O, CO_2_, SO_2_, and lesser amounts of H_2_S and CH_4_, along with N_2_ (a stable component of atmospheres of most planets of our solar system). Additional methane, CH_4_, and some hydrogen may have been available through serpentinization [24]. Methane, however, would have had a short lifetime in the upper atmosphere, being photochemically transformed with N_2_ into hydrogen cyanide, HCN, and cyanogen, NCCN, by short-wavelength UV-C photons [25,26]. H_2_O, CO_2_, SO_2_, HCN, and NCCN are, in fact, known precursors for the production of almost all the fundamental molecules of life, particularly under UV-C light [12,27,28,29,30,31,32].

### 2.2. Strong UV-C Photon Absorption and Rapid Excited State Dissipation of Fundamental Molecules

Organic molecules strongly absorb UV-vis light as a result of the conjugation of carbon covalent bonds (alternation of double and single carbon–carbon bonds—Figure 2), which gives rise to collective electron excitation. Conjugation requires the removal of protons from saturated hydrocarbons, giving carbon–carbon double bonds (C=C), therefore freeing electrons from their atomic orbitals to participate in molecular orbitals which can be either bonding or anti-bonding. The greater the conjugation number, the greater the wavelength of maximum absorption of the molecule (Figure 2). The energy differences between the ground state and first excited state for doubly and triply conjugated carbon bonded molecules fall within the range of energies of the soft UV-C photon spectrum incident at Earth’s surface during the Archean (Figure 3).

We have speculated [12] that the facility to form, and stability of, these UV-C light absorbing conjugations in carbon-based molecules are probably the principle reasons why Earth’s life is based on carbon and not, for example, on silicon, which has similar outer electron configuration but unstable conjugation in water [34], and thus unable to form chromophores.

**Figure 3 life-14-00912-f003:**
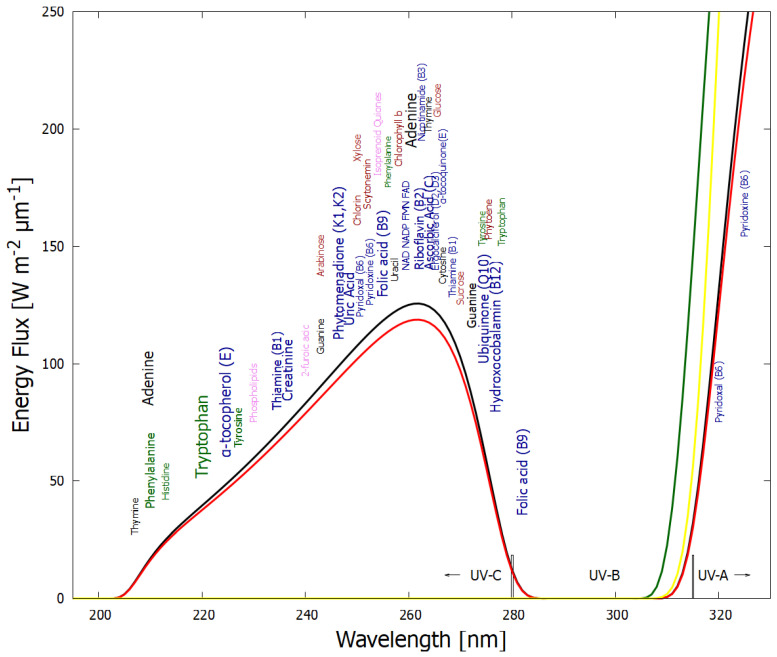
The spectrum of UV light available at Earth’s surface before the origin of life at approximately 3.9 Ga and until at least 2.9 Ga (curves black and red, respectively). This spectrum in the UV-C may even have persisted throughout the entire Archean until 2.5 Ga [35]. Atmospheric CO_2_, H_2_O, SO_2_ and probably some H_2_S, were responsible for the absorption of wavelengths shorter than ∼205 nm, and atmospheric gas aldehydes (e.g., formaldehyde and acetaldehyde, common photochemical products of CO_2_ and water) absorbed between about 280 and 310 nm [15,36]), approximately corresponding to the UV-B region. By around 2.2 Ga (green curve), UV-C light at Earth’s surface was completely extinguished by the pigments of oxygen and ozone resulting from organisms performing oxygenic photosynthesis. The yellow curve corresponds to the present surface spectrum. Energy fluxes are for the sun at the zenith. Over 50 of the fundamental molecules of life are plotted at their wavelengths of maximum absorption: nucleic acids (black), amino acids (green), fatty acids (violet), sugars (brown), vitamins, co-enzymes, and cofactors (blue), and pigments (red). We propose that these molecules were dissipatively structured as UV-C pigments under this light. The font size is roughly proportional to the relative size of the respective molar extinction coefficient of the pigment. Adapted with permission from [17]. 2015, K. Michaelian.

In Figure 3, approximately 50 fundamental molecules of life are plotted at their wavelength of maximum absorption on a plot of the incident Archean solar spectrum in the relevant UV-C, UV-B, and UV-A regions. There are very few fundamental molecules absorbing at wavelengths shorter than 205 nm since this hard UV-C region induces ionization and dissociation of carbon-based molecules. Post “late heavy bombardment” atmospheric (volcanic) gases CO_2_, H_2_O, SO_2_, and H_2_S absorbed very strongly here though dissociation. The sulfur isotope record suggests that between 3.2 Ga and 2.7 Ga, a thick organic haze blocked UV light in the 170–220 nm range due to photolysis of SO_2_ in the lower atmosphere [37]. Aldehydes (e.g., formaldehyde, CH_2_O) are a common photoproduct of this wavelength UV-C light on CO_2_ and H_2_O in the upper atmosphere [15]. These aldehydes then absorbed in the UV-B region through photo-dissociation into HCO, CO, H_2_, and H [15,36], therefore preventing this wavelength region from reaching Earth’s surface (Figure 3). Aldehydes are photochemical precursors of amino acids and ribose through Formose-like reactions [38,39]. Short-wavelength UV-C light (<205 nm) incident on the upper atmosphere (but not on the surface) thus also contributed to the dissipative structuring of the fundamental molecules at Earth’s surface.

Strong absorption persists over ∼±20 nm of the wavelength of maximum absorption for most fundamental molecules listed in Figure 3 (see, for example, adenosine, Figure 4). This broad ΔE absorption indicates, through the Heisenberg uncertainty relation (ΔEΔt≤ℏ/2), that these molecules dissipate very rapidly to the ground state and therefore must have a conical intersection [10] to rapid (small Δt) internal conversion (Figure 5), implying large molar extinction coefficients (Figure 4). The fundamental molecules of life, therefore, efficiently dissipated the entire UV-C light spectrum available on Earth’s surface during the Archean, and this was the thermodynamic (and, in fact, only) reason for their abiogenesis through dissipative structuring from inorganic material.

The photon-induced excitation of a molecule into an anti-bonding orbital (e.g., $\pi^{*}$) weakens the respective bond, decreasing the energy of the excited state upon elongation of the bond, leading to an intersection of the excited potential energy surface with the potential energy surface of the ground state (conical intersection—Figure 5). Conical intersection seams on the excited potential energy surface in multi-dimensional atomic coordinate space determine the internal conversion or photoisomerization and photoreaction products that can be reached after a photon absorption event. The direction and velocities of the approach of the nuclear coordinates to a conical intersection are important in defining the outcome [45].

An extended seam with different minimums can lead to different reaction products [46] such as those intermediate molecules on route to the photochemical synthesis of the nucleobase adenine (Figure 6). For the fundamental molecules of life, the photoexcited molecule reaches the conical intersection extremely fast (sub-picoseconds), implying that the conical intersection must be peaked (inverted cone-like on the excited state potential energy surface) and, overwhelmingly, only one reaction product is reached, the original ground-state configuration [47]. These photon-induced excited state energy dissipation processes are fundamental for understanding the phenomena of microscopic dissipative structuring [10] of the UV-C pigments (the fundamental molecules) discussed in the following section.

### 2.3. UV-C Photochemical Dissipative Structuring of the Fundamental Molecules

Details of the quantum photochemistry of organic molecules in electronic excited states can be found in Section 2 of our previously published paper on the dissipative structuring of adenine under UV-C light [12]. Here, we present only the main features.

The photochemistry of molecules is much richer than their ground-state thermal chemistry because (1) much larger energy barriers can be breached, (2) very endothermic reactions can occur, (3) anti-bonding orbitals can be reached, allowing reactions prohibited in the ground state, (4) triplet states can be reached from the electronically excited state, allowing intermediates that cannot be accessed through thermal reactions, and (5) electronically excited molecules are often converted into reactive radicals. Excited singlet states of carbon-based molecules have a richer chemistry than the spin “forbidden” excited triplet states.

Dissipative structuring of the carbon-based fundamental molecules at the origin of life occurred with photons of the long-wavelength UV-C (soft UV-C) and UV-B regions of the incident spectrum (Figure 3) because these had enough energy to transform carbon covalent molecules through isomerizations, additions, substitutions, rotation around double bonds, charge transfers, tautomerizations, etc., but not enough energy to significantly ionize the molecules and therefore dissociate them.

These transformations constituted the set of molecular manipulations available to the photochemical dissipative structuring of the fundamental molecules at the origin of life. They still occur today in many important photochemical structuring processes (e.g., photosynthesis) and other processes of life (e.g., vision) albeit now in the near UV or visible regions of the spectrum and through more complex biosynthetic pathways, often requiring ATP as an intermediate.

In Figure 6, the example of the dissipative structuring of the nucleobase adenine from 5 precursor molecules of HCN in water under soft UV-C light is given (see reference [12] for details).

In Figure 7, the wavelength of maximum absorption and the corresponding molar extinction coefficient of each intermediate molecule on route to adenine (Figure 6) is plotted on the graph of the surface UV-C solar spectrum during the Archean. A general tendency is observed towards absorption at greater photon intensities and towards greater molar extinction coefficients, a hallmark of microscopic dissipative structuring [10].

The rate of the dissipative structuring process as it proceeds from left to right in Figure 6 is dependent on the intensities of the incident spectrum at the wavelengths of maximum absorption of the intermediate molecules and on the photochemical quantum efficiencies for the photochemical reaction (dependent on the forward and backward rates over a potential energy barrier. These rates depend on the competing quantum efficiency for internal conversion to the ground state, $q^{IC}$, for the intermediate molecule—see Figure 8). The greater the quantum efficiency for internal conversion (dissipation), the greater the “force” for the evolution of the molecular concentration profile distribution towards the right in Figure 6 (see Figure 8 for explanation).

In reference [12], we have simulated the photochemical and chemical reactions with diffusion for this dissipative structuring of adenine occurring within a lipid vesicle starting with a plausible HCN concentration of $6.0\times 10^{-5}$ M in water under the probable surface light spectrum of the Archean given in Figure 7. Concentrations of adenine as high as $1.5\times 10^{-5}$ M are obtained after only 30 Archean days for probable ocean surface environmental conditions [12]. As detailed in Figure 8, the evolution towards adenine is driven by the increase of photon dissipation, which is related to entropy production.

Because the final products of photochemical dissipative structuring (e.g., the fundamental molecules of life) have a peaked conical intersection to internal conversion, allowing them to dissipate the electronic excitation energy to the ground state extremely rapidly, they become the *final* and *photo-stable* product of dissipative structuring in the relevant region of the solar spectrum.

It has been suggested [15,52] that the rapid (sub-picosecond) de-excitation of the excited nucleobases through their conical intersections had evolutionary utility in providing stability under the high flux of UV photons of the Archean since a peaked conical intersection reduces the lifetime of the excited state to such a degree that further chemical transformations are improbable. However, the quantum efficiency for internal conversion is never 100%. Photochemical reactions under UV-C light still occur for the fundamental molecules of life, particularly after excitation to the long-lived triplet state (e.g., cyclobutane pyrimidine dimers, 6–4, and other photoproducts, in RNA and DNA [53]). An apparently more congenial solution at nature’s disposal for avoiding radiation damage and degradation of biological function is transparency to, or reflectivity of, the offending UV light (e.g., saturated hydrocarbons or molecules based on silicon lacking the possibility for stable conjugation required for photon absorption—see Section 2.2). Our contention, however, is that a large antenna for maximum UV-C photon absorption and a peaked conical intersection for its rapid dissipation into heat are what would be expected as the “design goals” of thermodynamic dissipative structuring.

## 3. From Pigments to the Biosphere

### 3.1. Molecular Complexation through Natural Thermodynamic Selection on Dissipation

As with the nucleobase adenine, a plausible photochemical dissipative structuring route to guanine from the precursors hydrogen cyanide HCN and cyanogen NCCN (Section 2.1) in water under UV-C light has been found [32]. The molecular concentration profile evolves towards a similarly large molar absorption coefficient with conical intersections to rapid internal conversion. The same appears to be true for the pyrimidines, cytosine, thymine, and uracil, as well as for the fatty acids [31].

Traditional proposals for the abiogenesis of ribose and deoxyribose include formose-like reactions starting with aldehyde precursors under either high temperatures [54] or UV-C light [55] or through vacuum UV-C on extraterrestrial ices delivered to Earth in meteorites and comets [56,57]. Consistent with the theme of this paper, however, here we propose that ribose was dissipatively structured as a UV-C pigment from the precursor formaldehyde CH_2_O on the ocean surface under soft UV-C light. Formaldehyde itself is a product of UV-C photochemistry on an early Earth atmosphere [15,58] or UV-C photoredox chemistry on metal cyanides with HCN [59,60]. The steps involved in the UV-C dissipative structuring of ribose may have been (i) UV-C induced dimerization of formaldehyde to give glycoaldehyde, which then acts as a catalyst for further formaldehyde dimerization [61], (ii) UV-C-induced telomerization of two glycoaldehydes in water to produce malondialdehyde [62], (iii) tautmerization to the conjugated form of malondialdehyde, (v) reaction with an additional glycoaldeyde to form the conjugated form of pentanal. Alternatively, or additionally, a route to malondialdehyde could have been through UV-C light on fatty acids [63], which are themselves dissipative structures when in conjugated form [31].

An unsolved mystery in the origin of life is why ribose or deoxyribose would have joined to the nucleobases to form the nucleosides. One proposal is that, instead of the nucleobase and ribose forming separately, the nucleosides were built in a one-pot process [64]. However, there is no thermodynamic imperative for this seemingly contrived scenario, suggesting it would not occur naturally with important probability. Instead, our UV-C dissipative structuring theory suggests that two molecules will physically associate under a continuous UV-C flux if the overall photon dissipative efficacy (quantum efficiency for internal conversion) of the union increases (see, for example, Figure 8).

In step (vi) of the dissipative structuring of ribose proposed above, the pentanol linear structure converts to the 5-membered ring structure of ribose, and one of the double bonds is lost, reducing the conjugation by one, and thus the wavelength of maximum absorption decreases to 185 nm (Figure 2) with a tail extending to ∼214 nm. The attachment of ribose to the nucleobases confers three advantages to dissipation. First, it provides faster de-excitation through internal conversion using the conical intersection of ribose rather than that of the nucleobases [56]. This is affected by Förster energy transfer from the excited nucleobase to ribose. The net result is that the nucleoside complex (base plus ribose) has a larger oscillator strength or photon absorption compared to the nucleobase alone [65]. Second, the nucleobase becomes protected from possible ionization from occasional photons with energies greater than its ionization energy because ribose has a larger electron ionization energy [66]—Figure 9. Third, once both ribose and phosphate attach to the base, the resulting nucleotides polymerize and, therefore, become resistant to hydrolysis [67] and can serve as a scaffold for attachment of antenna molecules, such as the aromatic amino acid tryptophan [68] (see below). Ribose attaches readily to the nucleobases in experiments under UV-C light [69].

UV-C induced phosphorylation of the nucleosides would probably have occurred [70], formamide being a possible catalyst [71], given the large amount of phosphite detected in sediments before 3.5 Ga [72]. Nucleotides thus formed would have stacked through the π−π interaction between their aromatic rings and promoted by UV-C into phosphodiester-bonded stable dissipative RNA and DNA single-strand oligos.

As the sea surface temperature slowly dropped throughout the Archean, certain complementary short single-strand oligos would hydrogen bond, forming double strands. When the surface temperature fell below their denaturing temperature, a process of thermodynamic selection on only those double-strand oligos that absorb and dissipate photons most efficiently would have occurred [68]. This would happen through the mechanism of photon-induced denaturing, a phenomenon we have measured in the laboratory for DNA [40]. Those double strands that absorbed more photons have a higher probability of denaturing. This then would allow for the possibility of complementary oligo extension at colder overnight temperatures facilitated by magnesium- or iron-ions [70,73]. We call this reproduction and thermodynamic selection mechanism “ultraviolet and temperature assisted replication” (UVTAR), which would favor the most highly photon dissipative oligos (including possible stereochemically attached antenna amino acids [68], as explained in the following).

Similar increases in photon dissipation through other molecular associations and ensuing increases in complexity would have occurred naturally. For example, Yarus [74,75] has shown that tryptophan (Trp) has a chemical affinity to its RNA and DNA codons (e.g., TGG). Tryptophan has strong UV-C absorption with peaks at 220 and 278 nm (Figure 3) but no conical intersection to internal conversion and instead has a large quantum efficiency for fluorescence [76]. Given the chemical affinity of tryptophan to its codons through the π-ion interaction (or the π stacking interaction), once within the Förster range (1–10 nm), it could transfer its electronic excitation energy to one of the oligo bases through non-radiative dipole-dipole coupling and therefore use the conical intersection of the base to dissipate its excitation energy rapidly into heat. Tryptophan may thus have initially been a UV-C antenna donor molecule to RNA or DNA (Figure 10), giving the complex greater photon dissipative efficacy than the molecules acting independently. The codon with chemical affinity for tryptophan would thus become “programmed” into following generations of RNA or DNA since greater photon dissipation implies greater denaturing through the UVTAR enzymeless photon-induced denaturing [40] and therefore greater replication. This is another example of a mechanism giving rise to what we call “natural thermodynamic dissipative selection”.

The amino acid tryptophan enhances DNA stability under a high flux of UV-C light through intercalation between the bases of the macromolecule without significantly affecting its tertiary structure [77]. Intercalating molecules increase the rigidity of RNA and DNA strands, inhibiting cyclicization, which hinders enzymeless extension [78]. The efficiency of enzymeless extension in the presence of intercalating molecules, in fact, increases by various orders of magnitude [79]. L-tryptophan attachment also increases the circular dichroism of right-handed DNA [77], leading to a much more rapid procurement of homochirality through sea surface temperature morning-afternoon asymmetry and the temperature dependence of photon-induced denaturing [80]. A rapid procurement of homochirality would increase RNA or DNA potential for reproduction since a racemic distribution of nucleotides frustrates extension [81,82].

Not only tryptophan but other aromatic (tyrosine, phenylalanine, histidine) and non-aromatic amino acids that enhanced photon dissipation (e.g., amphiphilic amino acids keeping RNA and DNA at the ocean surface where incident UV-C light is most intense) would also have become programmed into RNA and DNA through this dissipative selection [68,83]. This would have led to the stereochemical era proposed by Yarus et al. [74,75] and to the incorporation of the first information into DNA, information useful for improving photon dissipation [68]. This dissipative selection operating during the stereochemical era most likely provided the basis of the specificity between amino acids and codons [68], an unresolved issue since 1967 when Carl Woese first emphasized the need for an explanation [84].

### 3.2. Organic Pigments in the Environment

Over Earth’s evolutionary history, organic pigments have appeared, covering an ever-increasing portion of the intense region of the solar spectrum [85]. In particular, pigments have evolved to absorb in the wavelength region where absorption of water is weak, principally between 205 and 800 nm. Stomp et al. [86] have demonstrated how neatly organic pigments fill even small photon niches left by water over all incident wavelengths, from the UV to the infrared.

Photosynthetic organisms today produce several different classes of pigments: chlorophylls, phycobiliproteins, flavonoids, carotenoids, and mycosporine-like amino acids (MAAs) covering the entire intense part of the present-day surface spectrum [87]. Cyanobacteria and plants actively regulate their pigment concentration profiles to best absorb their particular light environment [88].

Many pigments are known to have little or no effect on photosynthesis. For example, the carotenoids in plants that absorb between 400 and 475 nm, or the MAAs found in phytoplankton which display strong UV absorption maximum and high molar extinction between 310 and 360 nm [89]. MAAs have been assigned a UV photo-protective role, but this appears dubious since, in some cases, more than 20 MAAs have been found in the same organism, each with a different but overlapping absorption spectrum, determined by the particular molecular side chains [89]. If their principle function were photo-protective, then their existence in a particular plant or phytoplankton would be confined to those particular UV wavelengths that cause damage to the photosynthetic system and not to the whole broadband spectrum. It is particularly notable that the absorption spectrum of red algae, for example, has little correspondence to its photosynthetic activation spectrum [90].

Furthermore, there exist complex mechanisms that have evolved in plants to dissipate directly into heat photons absorbed on chlorophyll, bypassing photosynthesis completely. These mechanisms come in a number of distinct classes and operate by inducing the de-excitation of chlorophyll using a dedicated enzyme [85]. Over-wintering evergreen needles produce little photosynthesis due to the extreme cold but continue transpiring by absorbing photons and degrading these into heat through non-photochemical de-excitation of chlorophyll. Hitherto, these mechanisms were considered to be “safety valves” for photosynthesis, supposedly protecting the photosynthetic apparatus against light-induced damage [91]. However, their existence should now be attributed to thermodynamic mechanisms designed to augment the entropy production potential of a plant by increasing photon absorption, dissipation, and transpiration rates.

A global indication that biology is indeed fulfilling this fundamental thermodynamic role of dissipating photons is the lower albedo of regions with life compared to regions devoid of life. For example, the visible albedo of deciduous forests is 0.15 to 0.18, and that of coniferous forests is 0.09 to 0.15, while that of sandy deserts is about 0.30 [92], and that of rocky deserts (Gobi) is about 0.21 [93]. This is also true at wavelengths beyond the red edge (>700 nm) where forest albedo increases to about 0.3 [94], while sand and rocky desert albedo increases to about 0.50 [94,95]. The albedo of water bodies is also reduced by a concentrated surface microlayer of cyanobacteria [96], particularly for shallow incident angles and towards the ultraviolet where the albedo of pure water becomes large [97], and entropy production on photon dissipation is large.

Absorption or reflection (albedo) does not, however, encompass the whole story concerning the entropy production due to light interacting with material. How light is re-emitted from the material, related to a characteristic known as its wavelength-dependent emissivity, is also important. In reference [22], we show that, given a particular (non-zero) average albedo and emissivity, greater entropy production occurs when absorption is strongest at short wavelengths and emission is strongest at long wavelengths. Maximum entropy production occurs when the material acts as if it were a black body, i.e., with maximal absorptivity (zero albedo) and maximal emissivity (100%) across all wavelengths. Detailed calculations show that biological material more closely approximates a black body than non-biological material [22] and thus produces greater photon dissipation and entropy production.

Cyanobacteria not only produce pigments but also liberate these into their environment [85], a behavior incongruous with Darwinian theory. For example, the mycosporine-like amino acids (MAAs) and scytonemins are actively secreted into the surrounding water during cyanobacterial surface blooms [88,98,99,100]. These would thus seem to have the same function as the other bio-pigments in nature, acting as catalysts for the dissipation of photons into heat at Earth’s surface and the coupling of this heat to other abiotic entropy-producing processes, such as the water cycle, hurricanes, water and wind currents, etc. [85].

Pigment dissipative efficacy would have increased over time by (1) increasing molar extinction coefficients at shorter wavelengths, (2) maximizing emissivity at longer wavelengths, (3) increasing their quantum efficiency for de-excitation to the ground state through a conical intersection (internal conversion), (4) reducing their physical size, (5) exudation of pigments into the environment, and (6) inventing mechanisms (e.g., animals) to aid in pigment production and dispersal over the whole of Earth’s surface (Section 3.3.1).

### 3.3. Animals

#### 3.3.1. Dispersal of Pigments—The
Thermodynamic Role of Animals

Some 500 million years after the origin of the UV-C primordial pigment world, complex protein folded structures began to be dissipatively structured after the invention of oxygenic photosynthesis and the local, and then global, ozone UV shielding of the delicate folded protein structures. The short-wavelength UV-C and UV-B dissipation thus gradually became ever more relegated to ozone accumulating in the upper atmosphere, while dissipation on Earth’s surface evolved into the complex visible wavelength dissipative plants and animals of today’s ecosystems.

From the Darwinian perspective, emphasizing a struggle for existence, the proposition that free energy is not being efficiently utilized by life for its maintenance and reproduction but instead rapidly dissipated into heat without extraction of work appears wasteful and contradictory. Indeed, examples can be given where efficient utilization of free energy seems to be the fundamental organizational principle of certain biology. In fact, theories for ecosystem organization, including Darwinian theory, are based, at least partially, on the optimization of the utilization and storing of free energy [101]. Careful consideration, however, reveals that this optimization principle for the utilization of free energy is applicable only to animals and only approximately, but not to plants or other phototrophic organisms such as cyanobacteria.

Animals compose less than 5% of the biomass of the biosphere [102] and do not absorb and dissipate significant quantities of photons. They do, however, absorb and dissipate free energy in the form of chemical potential obtained by consuming plants and cyanobacteria. This obviously removes some potential for photon dissipation from the biosphere, and this is the thermodynamic origin of the “struggle for existence” among animals. Overall, however, animals actually increase photon dissipation since they provide many auxiliary services to the plants and cyanobacteria, from supplying nutrients and spreading seeds to pollination, therefore allowing organic pigments to cover the entire surface of Earth, particularly for areas with a scarcity of such important nutrients as water, phosphorous and nitrogen. Since most of the organic biomass is associated with pigment support, optimization of photon dissipation is consistent with optimization of biomass, a principle for which empirical evidence exists [101].

Because of their intricate root system, which allows the plants to draw up water from great depths for transpiration to foment the water cycle, plants are not mobile and depend on insects and other animals for their supply of nutrients, cross-fertilization, and seed dispersal into new environments. Burrowing rodents facilitate the delivery of bacterial reduced organic nutrients to the surface plants. The mobility and the short life span of many insects and animals with respect to that of plants mean that, through excrement and death, they provide a reliable mechanism for the dispersal of nutrients and seeds.

Zooplankton, crustaceans, and animal marine life in water perform a similar function as insect and animal life on land. These more mobile forms of life distribute nutrients throughout the ocean surface through excrement and death. Most of the ocean would be a pigment-less water desert if not for animals. It is noteworthy that dead fish and sea mammals do not sink rapidly to the bottom of the sea or lake but remain floating for considerable time on the surface where, as on land, bacteria break down the organism into its molecular components, allowing photon-dissipating phytoplankton to reuse the nutrients, particularly phosphorous and nitrogen. It is relevant that many algae blooms produce a neurotoxin with apparently no other end than to kill higher marine life [103]. There is also continual cycling of nutrients from the depths of the ocean to the surface, as deep diving mammals preying on bottom feeders release nutrients at the surface through excrement and death. Because of this animal-powered nutrient cycling, a much larger area of the ocean surface is rendered suitable for phytoplankton growth, leading to a greater surface absorption of sunlight and the catalysis of the water cycle than would otherwise be the case [96].

Animals thus provide a specialized gardening service to the plants and cyanobacteria, catalyzing their absorption and dissipation of sunlight in the presence of water, promoting photon dissipation, the water cycle, and entropy production. Strong empirical evidence suggests that ecosystem complexity, in terms of species diversity, is correlated with evapotranspiration [104]. Ecosystem species population profiles are selected based on local photon-dissipation efficacy since basins of attraction of the stationary states in population space are larger for those autocatalytic animal plus cyanobacteria/plant population profiles with higher net photon-dissipation [6]. This is similar to the way that the fundamental molecular pigment concentration profiles were selected at the origin of life based on photon dissipation efficacy (see Section 2.3). Animals thus obtain thermodynamic relevance (in fact, their fundamental relevance) from their ability to increase the plant and phytoplankton potential for photon dissipation and evaporation of water.

#### 3.3.2. The Survival Instinct

The survival instinct, apparently innate to all animals, is crucial for comprehending Darwinian theory. However, the theory provides no explanation of how it arises, only arguing that the organism would cease to exist if the survival instinct were not present. However, this is not an explanation of the origin of an innate characteristic, rather only the specification of an axiom of the theory. The physical origin of the survival instinct is, however, relevant to a deeper understanding of animal evolution.

To understand the physical/chemical origin of the survival instinct from within our dissipative thermodynamic paradigm, it is necessary to emphasize again how animals are involved in photon dissipation. As mentioned in Section 3.3.1, to be effective catalysts for the photon dissipative process occurring in phototrophic organisms, animals must reduce their consumption of chemical potential obtained by consuming these phototrophic organisms while, at the same time, increasing their efficiency for the dispersal of the nutrients for these. This gives rise to the “struggle for existence” observed among animals in nature and to a thermodynamic imperative for the survival instinct.

It is thus not only an individual struggle for survival, which is important in animal selection, but also the efficacy of nutrient dispersal. The spread of nutrients is affected by direct transportation (e.g., human farmers), urination, defecation, and death. The range of nutrient dispersal is proportional to the range of the animal wandering. In general, the larger the animal, the greater its range. Species with smaller geographical ranges are less effective dispersers and, in fact, observed to be more vulnerable to extinction [105]. Bringing settled nutrients up to the surface from deeper soil (e.g., rodent tilling) or ocean depths (e.g., bottom feeders) is also important. The latter is effective even far from shore, where few nutrients are found, to catalyze the replication of the photon-dissipating algae and cyanobacteria at the surface.

### 3.4. Coupling of Biotic and Abiotic Dissipative Processes in the Biosphere

Different irreversible processes can couple if their tensorial natures are compatible [1] and when this increases the dissipation of the imposed thermodynamic potential (generalized force). For example, the absorption and dissipation into the heat of photons in organic pigments in cyanobacteria on the ocean surface can increase evaporation by 8%, and absorption on pigments in plant leaves can increase evaporation from land surfaces by more than 200% [96]. This leads to a greater amount of water in the water cycle [106] and increased global photon-dissipation rates because (i) the heat of condensation at the cloud tops is emitted at a lower temperature than that of the ocean or leaf, surface, (ii) more clouds bring more water father inland, allowing dissipative organic pigments to establish themselves over a greater land area, (iii) photon emission into a greater solid angle due to similar day and night temperatures as a result of the increase in atmospheric water vapor content attributable to life, and (iv) warm water in the atmosphere can spawn other abiotic dissipative processes such as hurricanes, convection and wind.

Ecosystems are evolving under this thermodynamic selection of greater photon dissipation and not an ill-defined Darwinian “adaptability”. Modern ecosystems are more efficient at dissipating sunlight than were ancient ecosystems. This is evident from (1) the appearance of new pigments over time covering ever more of the solar spectrum, tending to reduce short-wavelength albedo [17,22]; (2) the spread of life and pigments over the entire Earth surface and the increase in surface biomass over time [107]; (3) the increase in water vapor in the atmosphere [106], maintaining day and night temperatures similar, therefore increasing the solid angle of the Earth’s emitted radiation (effectively doubling this part of the entropy production); (4) the greater biodiversity of modern ecosystems, implying more complete free energy dissipation [108].

### 3.5. Ecosystems

In general, ecosystem animal hierarchy (e.g., predators) increases the range of nutrient dispersal. This is, plausibly, the thermodynamic imperative for the existence of the ecological pyramid. An example is the re-introduction of wolves into their historical homelands in Yellowstone National Park after uncontrolled hunting led to their extinction in 1926. On their re-introduction in 1995, these top predators kept the deer and elk populations always on the move, therefore preventing them from overgrazing and helping to spread the nutrients of excrement and dead carcasses over a much larger area. This led to a general greening of the park [109], i.e., greater pigment density coverage and resulting photon dissipation.

In tropical ecosystems, nutrients are dispersed more by insects and heavy rains and thus rely less on large animals. Boreal ecosystems and cold ocean water environments, on the other hand, are much more dependent on large animals because rainfall is less, and cold temperatures mean that bacterial degradation is more confined to warm animal guts.

The above are examples of how natural thermodynamic selection operates at the ecosystem level, bringing it to climax states (stationary thermodynamic states [6]) of generally greater photon dissipation. In such a multi-dimensional species space, many stationary states exist around local peaks in the photon dissipation or entropy production (Figure 11). Those stationary states with the greatest attraction basin in this space, often corresponding to auto- or cross-catalytic situations, are those most frequently observed in nature [13].

## 4. Conclusions

There is compelling evidence and arguments supporting the hypothesis of a ‘’Pigment World” arising from common inorganic precursor molecules through photochemical dissipative structuring to dissipate the UV-C surface light of the Archean. UV-C light was present on Earth’s surface from before the beginning of life and for at least one thousand million years thereafter. Individual photons had sufficient energy to transform molecular carbon covalent bonds but not enough energy to ionize and, therefore, disassociate the molecules. The “spontaneous” emergence of an irreversible process such as life involves an imposed generalized thermodynamic potential (UV-C light), and the dissipative structures (pigments, now known as the fundamental molecules of life) formed in the process to dissipate the imposed potential.

The free energy available to be dissipated in UV light of wavelength less than 300 nm arriving at Earth’s surface because of the lack of an oxygen and ozone layer was more than 1000 times greater than that of all other non-photon energy sources combined during the Archean [110].

The fundamental molecules of life share extraordinary pigment-like optical properties in the UV-C. The wavelength of maximum absorption of many of these molecules coincides with the predicted window in the Archean atmosphere (Figure 3). The maximum absorption wavelength of the fundamental molecules, determined by the conjugation number, can be tuned by a simple protonation or deprotonation event, decreasing or increasing, respectively, this wavelength by about 30 nm (Figure 2). This would have allowed the dissipative structures, pigments, to easily evolve towards dissipation of higher intensity light at longer (or shorter) wavelengths and, therefore, simply “adapt” to a changing surface spectrum dependent on atmospheric conditions. The stability of carbon conjugations but not silicon conjugations in water solvent and the necessity of conjugation for photon absorption, could explain why life is based on carbon and not silicon.

Many of the fundamental molecules of life are endowed with peaked conical intersections [10,45,111] giving them broadband absorption (Figure 4) and high quantum yield for internal conversion, i.e., extremely rapid (sub-picosecond) dissipation of the photon-induced electronic excitation energy through a conical intersection into vibrational energy of molecular, atomic coordinates (Figure 5), and finally dissipated into the surrounding water solvent.

Many photochemical routes from common and simple precursor molecules to the synthesis of nucleic acids [12,27,32], amino acids [112], fatty acids [31], sugars [55,113], and other pigments [17] have been identified at these UV-C wavelengths.

The rate of photon dissipation within the Archean UV-C window generally increases after each incremental transformation on route to synthesis of the fundamental molecule (Figure 6), a behavior strongly indicative of dissipative structuring in the non-linear, non-equilibrium thermodynamic regime [10,12,31].

Even minor variations (e.g., tautomerizations or methylations) of the fundamental molecules, which often, in fact, endows them with lower Gibb’s free energy, eliminates, or significantly reduces, their extraordinary photon absorption and dissipation properties [114].

The primordial pigment world in the UV-C has evolved into the one of today that strongly absorbs over the whole of the near UV and visible regions. The UV-C and UV-B dissipation has been relegated to the life-derived pigments oxygen and ozone in the stratosphere, allowing more delicate and complex dissipative structuring pathways involving protein folding to evolve under the higher intensity visible light (e.g., Van der Waals bonding in chloroplasts important for photosynthesis).

Organic pigments are neatly adapted to niches in the water absorption spectrum. Exudation of UV pigments into the water environment (mycosporine-like amino acids and scytonemins) and the atmosphere (oxygen, isoprenes, terpenes, and other volatile organic compounds) by many phototrophic organisms does not appear to have a plausible explanation [85,115] other than the thermodynamic one of increasing the broadband dissipation of the incident solar spectrum and also indirectly catalyzing the production of ozone and thus allowing UV sensitive dissipative structuring pathways to develop under visible light.

The struggle for existence among animals, crucial to understanding Darwinian theory, is really a struggle to optimize catalysis of the photon-dissipating pigments. This is correlated with the range of the animal and the efficacy of its nutrient distribution. Animal hierarchy, the ecological pyramid, facilitates increases in animal range and distribution of nutrients. The propagation and dispersal of organic pigments, catalyzed by animals, and the coupling of this with abiotic dissipative processes, such as the water cycle, culminated in the apex photon dissipative structure, today’s biosphere.

Future evolution can be expected towards generally greater biosphere dissipative efficacy through (i) increasing individual pigment photon absorption and dissipation efficiency, (ii) extending dissipation to longer wavelengths beyond the red edge with new pigments, (iii) increasing global areal pigment coverage, and (iv) stronger coupling of biotic and abiotic dissipative processes to further red-shift Earth’s emitted spectrum.

Life began as the dissipative structuring of a “UV-C pigment world” and has evolved into the highly dissipative “broadband pigment world” of today.

## Figures and Tables

**Figure 2 life-14-00912-f002:**
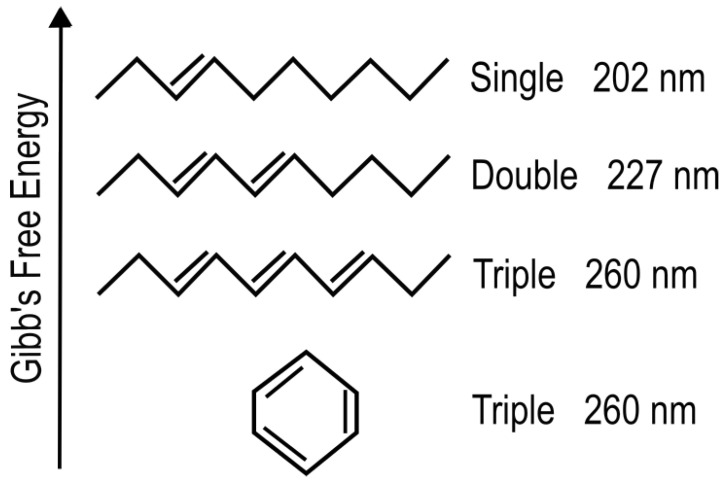
Conjugated carbon molecules are more stable (lower Gibb’s free energy in the ground state) as compared to saturated molecules, but, more importantly, provide new collective electron orbitals, giving rise to excited states at energies adequate for the absorption of soft UV-C photons. The greater the conjugation number, the greater the wavelength of maximum absorption. The wavelength of maximum absorption of the chromophore can, therefore, be tuned simply by a protonation or deprotonation event. Conjugation is also important for giving molecules a conical intersection (see below) allowing rapid dissipation of the electronic excited state energy into heat (internal conversion). Reprinted with permission from Ref. [33]. 2023, K. Michaelian.

**Figure 4 life-14-00912-f004:**
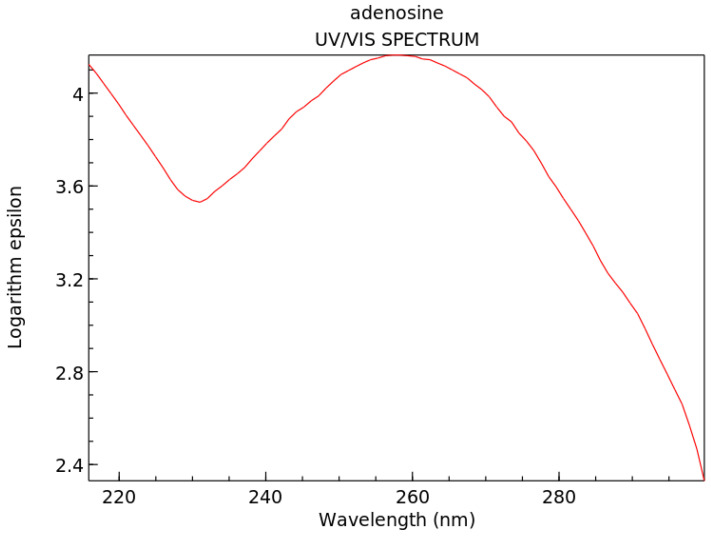
Absorption spectrum of adenosine in the soft UV-C region. The logarithm of molar extinction coefficient ϵ, in units of M^−1^cm^−1^, is plotted as a function of wavelength. Peak absorption at 260 nm corresponds to the peak in the incident UV-C spectrum arriving at Earth’s surface during the Archean (Figure 3). Large broadband absorption implies rapid (sub-picosecond) dissipation of the electronic excitation energy into heat through a conical intersection (Figure 5). Public domain NIST Chemistry WebBook (https://webbook.nist.gov/chemistry accessed on 1 June 2024).

**Figure 5 life-14-00912-f005:**
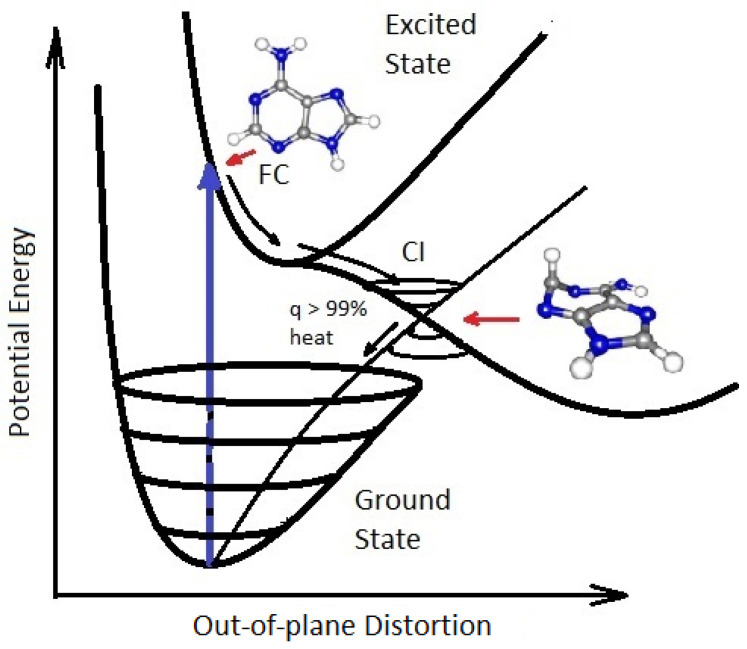
A conical Intersection (CI) for excited adenine showing a degeneracy of the electronic excited state with the vibrational states superimposed on the electronic ground state after a UV-C photon absorption event (blue arrow), which induces a nuclear coordinate deformation from the molecules original structure in the Franck-Condon (FC) region to activation of an N9–H stretch or a ring-puckering motion known as pyramidalization (shown in the diagram). The most probable deformation depends on the incident photon energy and protonation state. It is this deformation, resulting from the excitation of an anti-bonding state (e.g., $\pi \to \pi^{*}$), which leads to a lowering of the excited potential energy surface such that it intersects with vibrational states of the electronic ground state, resulting in the conical intersection. Conical intersections provide rapid (sub-picosecond) dissipation of the electronic excitation energy into vibrational energy (heat). The quantum efficiency, *q*, for this dissipative route is very large (>99%) for many of the fundamental molecules of life, making them photochemically stable and, more importantly for our theory, very efficient at UV-C photon dissipation. Another common form of coordinate transformation mediated through conical intersections is proton transfers within the molecule or with the solvent environment, and this may have relevance to enzymeless photon-induced denaturing of RNA and DNA [40] (see also Section 3.1). The diagram is based on data from Andrew Orr-Ewing [41], Roberts et al. [42], Kleinermanns et al. [43], and Barbatti et al. [44]). Reprinted with permission from Ref. [12]. 2021, K. Michaelian.

**Figure 6 life-14-00912-f006:**
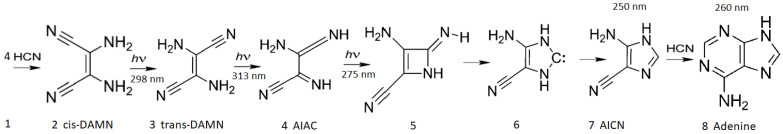
The photochemical dissipative structuring of adenine from 5 molecules of hydrogen cyanide (HCN) in water, first observed by Ferris and Orgel (1966) [27,30]. Four molecules of HCN (**1**) are transformed into the smallest stable oligomer (tetramer) of HCN, known as cis-2,3-diaminomaleonitrile (cis-DAMN) (**2**), which, under a constant UV-C photon flux, isomerizes into trans-DAMN (**3**) (also known as diaminofumaronitrile, DAFN) which can be converted, on absorbing two more UV-C photons, into an imidazole intermediate, 4-amino-1H-imidazole-5-carbonitrile (AICN) (**7**). Hot ground-state thermal reactions with another HCN molecule or its hydrolysis product formamide (or ammonium formate) lead to the purine adenine (**8**). This is a microscopic dissipative structuring process that ends in adenine [10,12], a UV-C pigment with a large molar extinction coefficient at the maximum intensity of the UV-C Archean surface solar spectrum (260 nm) and a peaked conical intersection facilitating rapid dissipation of photons at these wavelengths (Figure 3 and Figure 4). Reprinted with permission from Ref. [12]. 2021, K. Michaelian.

**Figure 7 life-14-00912-f007:**
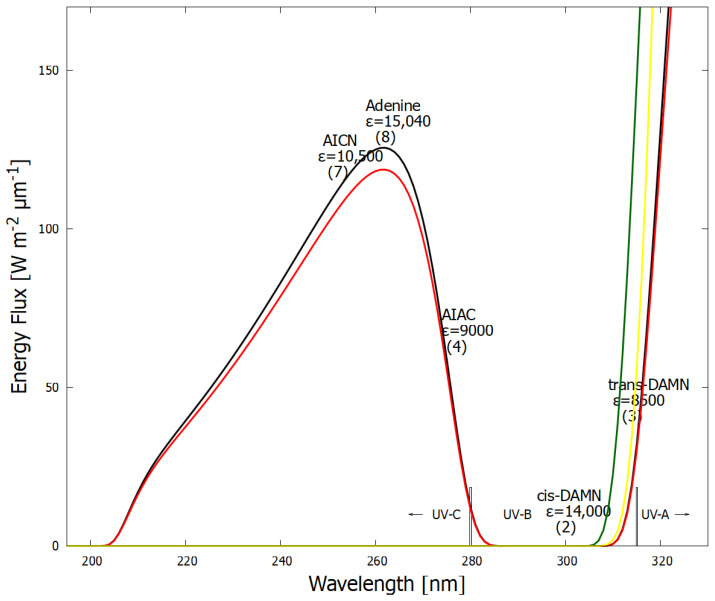
The wavelength of maximum absorption, and the molar extinction coefficient ϵ (units of M^−1^ cm^−1^) at that wavelength, of intermediate molecules on the way to the dissipative structuring of adenine (Figure 6) is plotted on the spectrum of UV light available at Earth’s surface before the origin of life at approximately 3.9 Ga and until at least 2.9 Ga (curves black and red respectively). The general tendency towards absorption at greater photon intensities and towards greater molar extinction coefficients are the hallmarks of dissipative structuring. This is a microscopic analogy of the macroscopic dissipative structuring and growth in size and dissipating strength of a hurricane as it steers itself over warmer ocean surface water, due in part to algal pigment concentration [48]. The wavelengths of maximum absorption and the molar extinction coefficients at those wavelengths were obtained from references [30,49,50,51].

**Figure 8 life-14-00912-f008:**
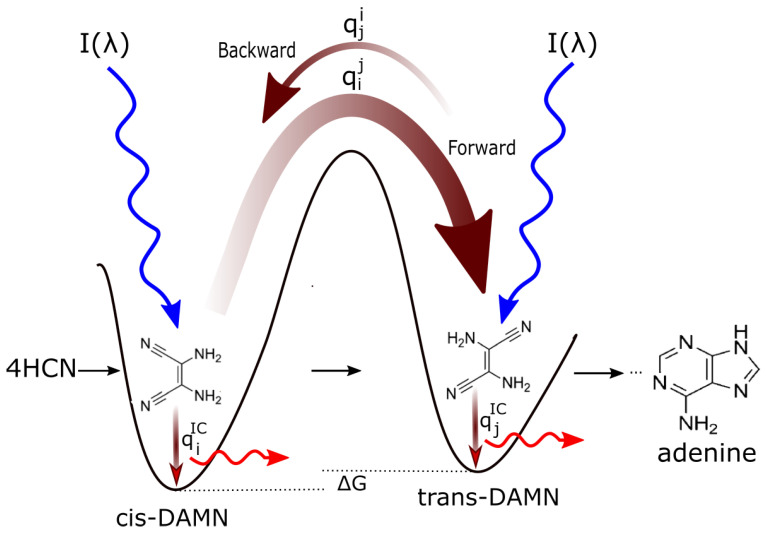
Mechanism of dissipative structuring for the evolution towards molecular structures of ever greater photon dissipative efficacy (microscopic dissipative structuring) on route to the fundamental molecules (in this case, the UV-C pigment adenine, see Figure 6). The high activation barriers between ground-state configurations mean that reactions will not proceed spontaneously but only on coupling to photon absorption events. Forward and backward rates depend on photon intensities I(λ) at the different wavelengths of absorption for the two structures and on the phase-space widths of paths on their excited potential energy surface leading to the conical intersection giving rise to the particular transformation, implying different quantum efficiencies for the forward ($q_{i}^{j}$) and backward ($q_{j}^{i}$) reactions. For example, assuming that the intensity of the incident spectrum is uniform over wavelength, and since $q_{i}^{j}+\cdots q_{i}^{IC}=1$ and $q_{j}^{i}+\cdots q_{j}^{IC}=1$ (where the “⋯” represents quantum efficiencies for other possible molecular transformations), those stationary states (corresponding macroscopically to molecular concentration profiles) with greater photon dissipative efficacy (higher quantum efficiency for internal conversion $q_{j}^{IC}$) will therefore gradually become predominant under a continuous UV-C photon flux (since, in general, $q_{j}^{i}<q_{i}^{j}$ if $q_{j}^{IC}>q_{i}^{IC}$), independently of the sign or size of the difference in the Gibb’s free energies ΔG of the molecules. This process of selection of molecular concentration profiles of ever greater photon dissipative efficacy, driving evolution towards the right in the diagram, we call *natural thermodynamic selection*. It is considered in a simulated reaction-diffusion system within a vesicle for the abiogenesis of adenine from HCN under UV-C light in our reference [12]. Reprinted with permission from Ref. [12]. 2021, K. Michaelian.

**Figure 9 life-14-00912-f009:**
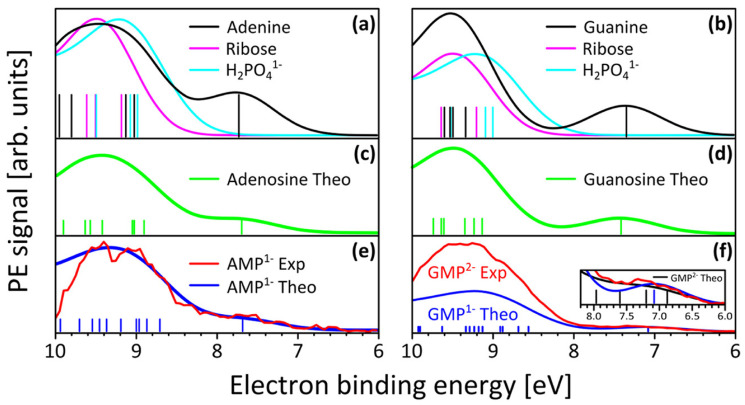
The ionization probability of adenine (**a**), guanine (**b**), and ribose as a function of energy compared to that of the nucleobase plus sugar complex, adenosine (**c**) and guanosine (**d**), and to adenosine monophosphate (**e**) and guanosine monophosphate (**f**). The thermodynamic dissipation theory argues that molecular complexation, such as a nucleobase attaching to ribose, will only occur if such an association leads to a system with greater photon dissipative efficacy. The conical intersection of ribose (and deoxyribose) to internal conversion is significantly more rapid than that of the nucleobase. By passing its photon-induced excitation energy onto ribose, the nucleoside complex becomes more efficient at dissipating than the nucleobase. Furthermore, the nucleobase is protected from ionization by hard UV-C (<205 nm, ∼6 eV) since, as shown in the figure, the ionization energy of ribose is larger than that of the base. The bases are also protected from hydrolysis by polymerization, which can happen on attachment to ribose and phosphate [67]. Together, the base plus sugar is a more efficient and more robust system for dissipating the Archean UV surface spectrum than the molecules separated. Reprinted with permission from Ref. [66]. 2015, American Chemical Society.

**Figure 10 life-14-00912-f010:**
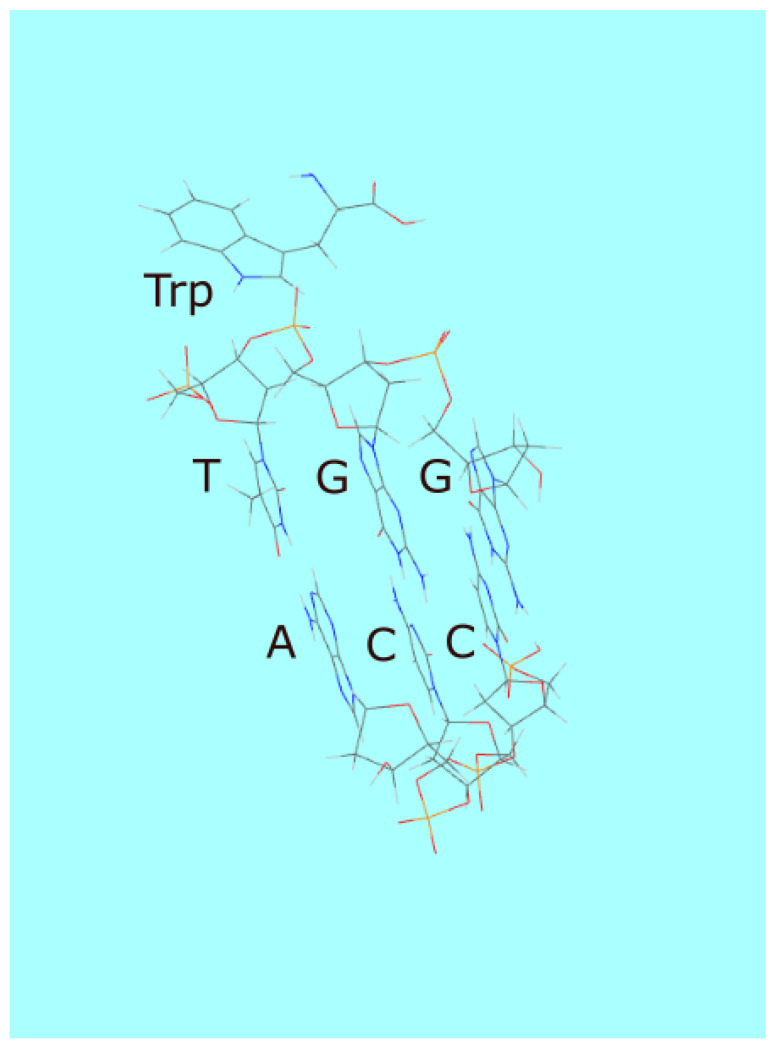
An example of increasing complexity through the dissipative structuring of an amino acid and DNA association, leading to greater photon dissipation. Tryptophan is attracted to its codon through a π-ion interaction (also π-stacking). Once within the Förster range of its DNA codon, it can transfer its photon-induced electronic excitation energy to one of the bases and, therefore, use the conical intersection of the base to dissipate its excitation energy rapidly into heat. Tryptophan may thus have initially been a UV-C antenna molecule for DNA, giving the complex greater photon dissipative efficacy than that of the two pigment molecules acting separately. This would have given the complex greater probability for UV-C induced denaturing and, therefore, reproduction, a process we refer to as “thermodynamic dissipative selection” [68].

**Figure 11 life-14-00912-f011:**
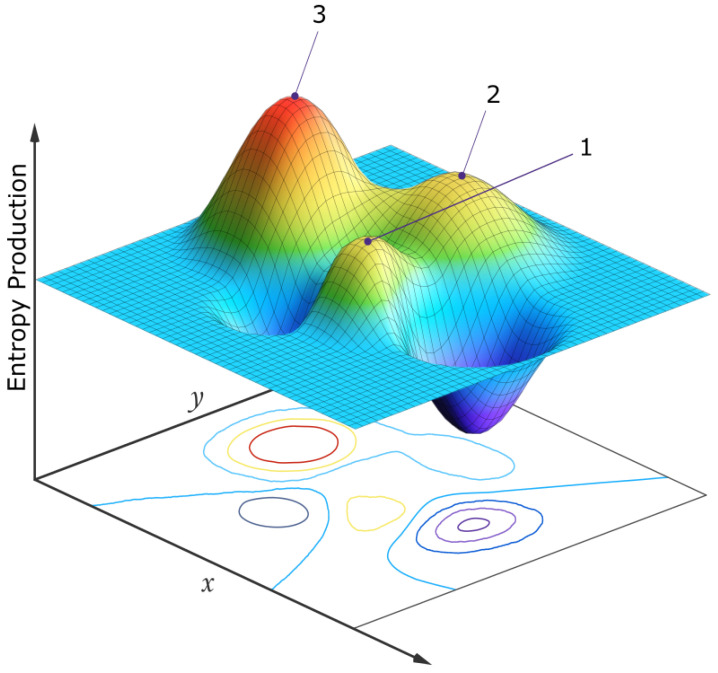
A simplified 2-dimensional schematic representation of the multi-dimensional entropy production surface (EPS) of a generalized phase space for a biosystem under a constant solar photon potential. The variables *x* and *y* at the origin of life may be, for example, the concentrations of different pigment molecules, while for an ecosystem of today, the variables may be the populations of different species. Three locally stable stationary states at local peaks in the entropy production surface are presented. On large enough external or internal perturbation, the system evolves from one stationary state to another. Although fluctuations are generally stochastic, the system will most often be found in those stationary states with a larger attraction basin, often corresponding to a higher peak in photon dissipation (e.g., the stationary state labeled “3”). Autocatalytic stationary states have higher peaks and larger attraction basins in this generalized phase space and are thus more probable. For molecules, this corresponds to concentration profiles with greater quantum efficiency for dissipation to the ground state through a conical intersection. For an ecosystem, this corresponds to animal and plant population profiles, giving greater total photon dissipation (time local climax ecosystems). If the system began in stationary state 1, its most probable future evolution would be 1→3, but any combination is possible. For the biosphere, the *x* and *y* variables might be the number of species in two different clades, and sub-peaks (not shown) corresponding to different species populations would exist on the main peaks, and evolution would usually be local, among the sub-peaks, but every once in a while a perturbation may be large enough (for example, an asteroid impact) to move the system from one main peak to another (e.g., 1→3, mammals y becoming more prominent than dinosaurs *x*). Point, cyclic, or even chaotic dynamics are allowed superimposed on these peaks [6]. The dimensionality of the generalized phase space is not fixed but evolves over time, providing new “shorter” routes to larger peaks of entropy production (e.g., the re-introduction of a population of wolves into the ecosystem of Yellow Stone National Park, see text). Reprinted with permission from Ref. [13]. 2022, K.Michaelian.

## Data Availability

No new data were created or analyzed in this study. Data sharing is not applicable to this article.

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
