# Peer review of "The Pigment World: Life’s Origins as Photon-Dissipating Pigments"

_life, 2024, doi:10.3390/life14070912_

Round 1

Reviewer 1 Report

Comments and Suggestions for Authors

The paper treats the interaction of UV with pigments with interest in dissipation and irreversibility.

 In my opinion, the paper is very interesting, but I must suggest some improvements:

 - It is not clear the aim of the paper: what does really the author aim?

 - It is not clear the conclusions

 - In the introduction the author develops some consideration on Prigogine's approach, but no analytical relations are used in the paper: I suggest considering some papers from Lucia, Grisolia, Sciubba, Markov, Bejan, Bustamante on irreversible thermodynamics of biosystems to improve the analytical approach

 - I suggest to introduce some information on the experimental setups considered in the analyses

 After these improvements, the paper could be considered for publication

Author Response

I thank the reviewer for their time and kind words, and for their useful comments that have helped improve the manuscript. 

In my opinion, the paper is very interesting, but I must suggest some improvements: 
- It is not clear the aim of the paper: what does really the author aim? 
- It is not clear the conclusions.

I have made the aim of the article explicit in the Introduction (line 93 - highlighted) and have improved the Conclusions. 

In the introduction the author develops some consideration on Prigogine's approach, but no analytical relations are used in the paper: I suggest considering some papers from Lucia, Grisolia, Sciubba, Markov, Bejan, Bustamante on irreversible thermodynamics of biosystems to improve the analytical approach.

I have included a new sentence beginning on line 33 (highlighted) acknowledging references to the analytic treatment by Prigogine, our own, and two new citations to relevant papers by Lucia (references 7 and 9).

I suggest to introduce some information on the experimental setups considered in the analyses.

I have included a new paragraph beginning on line 274 (highlighted) describing in more detail the photon dissipation dependent mechanism of denaturing and extension, and also cited the references to the relevant experiments by us and by others.

I have made numerous small corrections throughout the text to improve the redaction.

Reviewer 2 Report

Comments and Suggestions for Authors

UV light is known to have played a significant role on early Earth, promoting many transformations considered crucial to the emergence of biomolecule precursors. However, it is less understood why nature developed the essential molecules that are critical to the origin of life. In this manuscript, Karo Michaelian describes how the UV energy dissipation of molecules could have been the underlying reason for “natural selection,” leading to the accumulation of fundamental life molecules, and even RNA/DNA oligonucleotides. This thermodynamic dissipation theory explains the evolution of molecules and thus has significant implications for exploring the origin of life. Additionally, the author discusses the role of so-called pigments in environments and the catalytic role of animals in the propagation and dispersal of organic pigments, offering insights into the development of the biosphere. The manuscript effectively integrates concepts from physics, chemistry, and environmental science, indicating the interdisciplinary nature and generality of the proposed theory. The reviewer recommends publication in Life after addressing the following issues:

1. The title “The Pigment World” is too broad and does not accurately reflect the ideas conveyed in the article. A more detailed and precise title is recommended to avoid any misconceptions that might arise from the literal interpretation of "pigment."

2. The reviewer noted that several figures (Figures 2, 3, 5, 6, 7, 8, 11) have been used in another paper by the author (The Non-Equilibrium Thermodynamics of Natural Selection: From Molecules to the Biosphere, Entropy 2023, 25, 7, 1059), and there are sections of narrative content that are almost identical to the previous Entropy paper (lines 108-118, 145-154, 209-223). It is necessary to rewrite these sections to better suit the needs of this paper.

Author Response

I thank the reviewer for their time and kind words and for their useful comments that have helped improve the manuscript.

The title “The Pigment World” is too broad and does not accurately reflect the ideas conveyed in the article. A more detailed and precise title is recommended to avoid any misconceptions that might arise from the literal interpretation of "pigment."

The title of the manuscript has been changed to, “The Pigment World: A Photon Dissipative Perspective on the Origin and Evolution of Life”. 

The reviewer noted that several figures (Figures 2, 3, 5, 6, 7, 8,11) have been used in another paper by the author (The Non-Equilibrium Thermodynamics of Natural Selection: From Molecules to the Biosphere, Entropy 2023, 25, 7, 1059), and there are sections of narrative content that are almost identical to the previous Entropy paper (lines 108-118, 145-154, 209-223). It is necessary to rewrite these sections to better suit the needs of this paper.

I have now obtained permission and given proper reference to all figures that have been used before. The figures are essential for an understanding of the proposal. Figure 7 has not been used before. The sections mentioned by the reviewer have been rewritten. The relevant revised text corresponds to lines 112-120, 147-160, 211-223 in the new version and has been highlighted.

I have made numerous small corrections throughout the text to improve the redaction.

Round 2

Reviewer 1 Report

Comments and Suggestions for Authors

I thank the author to have addressed all my suggestions.

I suggest to accept the paper as it is.